# The Bacterial Amyloids Phenol Soluble Modulins from *Staphylococcus aureus* Catalyze Alpha-Synuclein Aggregation

**DOI:** 10.3390/ijms222111594

**Published:** 2021-10-27

**Authors:** Caroline Haikal, Lei Ortigosa-Pascual, Zahra Najarzadeh, Katja Bernfur, Alexander Svanbergsson, Daniel E. Otzen, Sara Linse, Jia-Yi Li

**Affiliations:** 1Neural Plasticity and Repair Unit, Wallenberg Neuroscience Center, Department of Experimental Medical Science, Lund University, 22184 Lund, Sweden; caroline.haikal@med.lu.se (C.H.); alexander.svanbergsson@med.lu.se (A.S.); 2Department of Biochemistry and Structural Biology, Lund University, 22100 Lund, Sweden; lei.ortigosa@biochemistry.lu.se (L.O.-P.); katja.bernfur@biochemistry.lu.se (K.B.); sara.linse@biochemistry.lu.se (S.L.); 3Interdisciplinary Nanoscience Centre (iNANO), Aarhus University, Gustav Wieds Vej 14, 8000 Aarhus C, Denmark; Zahra_najarzadeh@yahoo.com (Z.N.); dao@inano.au.dk (D.E.O.); 4Health Sciences Institute, China Medical University, Shenyang 110112, China

**Keywords:** alpha-synuclein, bacterial amyloids, aggregation, protein folding, Parkinson’s disease

## Abstract

Aggregated α-synuclein (α-syn) is the main constituent of Lewy bodies, which are a pathological hallmark of Parkinson’s disease (PD). Environmental factors are thought to be potential triggers capable of initiating the aggregation of the otherwise monomeric α-syn. Braak’s seminal work redirected attention to the intestine and recent reports of dysbiosis have highlighted the potential causative role of the microbiome in the initiation of pathology of PD. *Staphylococcus aureus* is a bacterium carried by 30–70% of the general population. It has been shown to produce functional amyloids, called phenol soluble modulins (PSMαs). Here, we studied the kinetics of α-syn aggregation under quiescent conditions in the presence or absence of four different PSMα peptides and observed a remarkable shortening of the lag phase in their presence. Whereas pure α-syn monomer did not aggregate up to 450 h after initiation of the experiment in neither neutral nor mildly acidic buffer, the addition of different PSMα peptides resulted in an almost immediate increase in the Thioflavin T (ThT) fluorescence. Despite similar peptide sequences, the different PSMα peptides displayed distinct effects on the kinetics of α-syn aggregation. Kinetic analyses of the data suggest that all four peptides catalyze α-syn aggregation through heterogeneous primary nucleation. The immunogold electron microscopic analyses showed that the aggregates were fibrillar and composed of α-syn. In addition of the co-aggregated materials to a cell model expressing the A53T α-syn variant fused to GFP was found to catalyze α-syn aggregation and phosphorylation in the cells. Our results provide evidence of a potential trigger of synucleinopathies and could have implications for the prevention of the diseases.

## 1. Introduction

Aggregated alpha-synuclein (α-syn) is a pathological hallmark of Parkinson’s disease (PD), multiple system atrophy and Lewy body dementia. The majority of synucleinopathies are of idiopathic nature and the environment is thought to play a causative role. In Braak’s landmark studies, α-syn inclusions were identified in the dorsal motor nucleus of the vagus at very early stages of PD. It was hypothesized that PD pathology could be initiated peripherally, either in the gastrointestinal or the olfactory system, where it would spread to the brain [1,2]. Several studies in rodents have shown that α-syn can propagate from the intestinal lumen, wall or enteric neurons to the brain [3,4,5,6]. Other studies have shown propagation of preformed α-syn fibrils from the olfactory bulb to the amygdala and entorhinal cortex [7,8,9]. For a deeper understanding of PD and other synucleinopathies, it is of interest to examine peripheral factors, which could trigger the aggregation of the otherwise soluble α-syn.

Both the olfactory cavities and the gastrointestinal tract harbor large numbers of microorganisms that are known to prime and modulate the immune system as well as directly modulate α-syn aggregation. Both LPS and bacterial chaperones have been shown to affect the kinetics of aggregation and toxicity of α-syn [10,11,12]. Cross-seeding of α-syn has previously been demonstrated not only with human amyloid proteins (tau [13] and Aβ42 [14]) but also with amyloid proteins of bacterial origin (Curli from *E. coli*) [11,15]. Several bacteria and fungi produce amyloids, which often help in the maintenance of a biofilm and adherence [16]. One such bacterium is the gram-positive *Staphylococcus aureus*.

*S. aureus* is the bacterium responsible for the majority of skin and soft-tissue infections [17] as well as a common skin, nose and even GI commensal [18,19,20,21]. *S. aureus* produces amyloid proteins called phenol soluble modulins (PSMs), which participate in biofilm formation. The α type of the PSMs comprise four different hydrophobic 20–25 amino-acid long peptides, often formylated at the N-terminal. The pH modulation of their aggregation mechanisms was recently reported (41) with PSMα1 found to aggregate through a mechanism dominated by secondary nucleation at neutral pH. Other factors also modulate PSM fibrillation, such as the sulfated polysaccharide heparin [22]. The production of PSMα peptides by *S. aureus* is stringently regulated and is associated with virulence. PSMα peptides are secreted by *S. aureus* at high concentrations, with high nanomolar concentrations acting as chemoattractants for neutrophils and micromolar concentrations causing cytolysis [23,24]. Their expression has been shown to be upregulated upon *S. aureus* phagocytosis by neutrophils promoting phagosome lysis and bacterial escape [25,26]. PSMα peptides can rapidly induce neutrophil extracellular trap (NET) formation [24]. Interestingly, NETs have been shown to co-localize with amyloids in human tissues [27].

PSMα peptides produced by *S. aureus* have been shown to directly activate sensory neurons of the skin and increase firing in cultured dorsal root ganglia (DRG) neurons by disrupting membranes and forming pores, allowing cation influx and depolarization [28,29]. In contrast, they have been shown to inhibit the firing of sensory neurons in the enteric nervous system leading to modulation of the secretion and motility of the intestine [30]. Nociceptive hypersensitivity and GI disturbances have been described both in PD patients and PD rodent models. Skin biopsies have revealed small fiber neuropathies or mixed fiber polyneuropathy in PD patients [31] and Braak has described α-syn deposits in neurons of the lamina I of the spinal cord, which directly project to the thalamus from peripheral nociceptive neurons [32]. Lewy body pathology has also been described in the enteric nervous system (ENS) [33] and neurons of the DRG [34]. PSMα peptides could as such interact with α-syn in peripheral sensory neurons.

In the present study, we hypothesized that these PSMα peptides could modulate α-syn aggregation. We monitored the aggregation kinetics of α-syn in vitro upon the addition of different PSMαs at a range of concentrations. The PSMαs induced rapid aggregation of monomeric α-syn into fibrils, which could induce seeding and phosphorylation in human embryonic kidney (HEK) cells expressing the A53T variant of α-syn fused to green fluorescent protein (GFP).

## 2. Results

### 2.1. Phenol Soluble Modulins Catalyze α-Syn Aggregation

We investigated whether the PSMαs could induce α-syn aggregation at physiologically relevant concentrations, i.e., in the 1–100 µM range. To study the effects on the kinetics under conditions where secondary nucleation strongly dominates over primary nucleation (pH 5.5 and low ionic strength), or conditions where secondary nucleation is less dominant (pH 7.5 and moderate ionic strength), the experiments were repeated in two different buffers. The PSMα peptides may interact with α-syn in the cytoplasm of sensory neurons or in the extracellular space, which further motivates the inclusion of the neutral pH condition.

We first monitored the aggregation kinetics of 25 μM α-syn under quiescent conditions in the presence or absence of different PSMα peptides in 10 mM Tris, 50 mM NaCl, pH 7.6. The addition of each of the four PSMα peptides induced a rapid increase in ThT fluorescence, which was not observed for α-syn incubated on its own or with DMSO at the same concentrations as used with the peptides (Figure 1 and Appendix A). All four PSMα peptides significantly shortened the lag phase for α-syn aggregation to less than 20 h, except at the lowest concentrations of PSMα2 and PSMα3. The ThT curves plateaued before 100 h, except in the presence of the highest concentrations of PSMα1 and PSMα4 (Figure 1 and Appendix A). The concentration dependence was monotonic for PSMα2 and PSMα3, with the strongest catalytic effect at the highest peptide concentration tested (78 µM), and biphasic for PSMα1 and PSMα4 with the strongest catalytic effect observed at 3–9 µM. The results indicate that PSMα peptides potently catalyze α-syn aggregation in buffers mimicking physiological conditions, rapidly increasing the ThT fluorescence intensity even at low peptide concentrations. The concentration-dependence variance between the similarly sized and charged peptides suggests that this catalysis is sequence-specific.

In mildly acidic buffer, 10 mM MES, pH 5.5 with no additional salt, all four PSMα peptides significantly shortened the lag phase for α-syn aggregation to less than 20 h, and the ThT curves plateaued before 100 h even at low μM PSMα concentrations (Appendix A). At the examined concentrations, the overall aggregation was slower at pH 7.5 compared to 5.5.

To verify the catalytic effect of the peptides on α-syn aggregation, additional experiments were performed at higher α-syn monomer concentrations, at higher pH (7.4) and higher salt concentration (150 mM) in the presence of polystyrene surfaces, known to catalyze primary nucleation of α-syn. The PSMα peptides were again shown to accelerate the increase in ThT fluorescence (Appendix A).

### 2.2. PSMα Modulation of α-Syn Aggregation Is DMSO Independent

As the PSMα peptides were dissolved in DMSO, the samples with the peptide also contained DMSO, the concentration of which increased with the peptide concentration. As the PSMα-α-syn aggregated samples would later be added to cells, the lower concentrations of the peptides were not supplemented with additional DMSO. However, in one control experiment, we supplemented the different PSMα2 peptide dilutions with DMSO to correspond to the highest DMSO concentration (8.9% *v*/*v*, equal to 1.25 M) used in the previous experiments. The effects of the PSMα2 concentrations with additional DMSO on α-syn were similarly examined. The kinetic curves obtained with this PSMα2 dilutions series show similar trends as the PSMα2 dilutions without supplemented DMSO (varying DMSO concentrations) (Appendix A).

To ensure that the effects on α-syn aggregation observed were due to the peptides, and not to DMSO or impurities, one aliquot of PSMα2 was purified with size exclusion chromatography before the kinetics experiments. Different fractions of the central peak were added to freshly purified α-syn monomer in 10 mM Tris, 50 mM NaCl, pH 7.6 and compared to PSMα2 dissolved in DMSO. The results indicate similar effects of the purified PSMα2 fractions (without any DMSO) compared to the original DMSO-dissolved peptide (Appendix A).

### 2.3. PSMα Peptides Modulate α-Syn Aggregation Differently

Next, we wanted to elucidate the mechanisms underlying the PSMα-catalyzed α-syn aggregation. As a first step, we examined the effects of the different PSMα concentrations on the rate of α-syn aggregation. The time at which ThT intensity reached half the intensity of the plateau was calculated for each concentration of the four different PSMα peptides and plotted against the concentration of the peptides (Appendix A). At pH 7.6, the double logarithmic plots of PSMα1 and PSMα4 indicate an optimal catalytic concentration, above and below which the lag phase is prolonged. In contrast, PSMα2 and PSMα3 exhibit linear curves, or curves that flatten out, with negative slopes, indicating a reduction of half-times as a function of concentration. This is interesting as PSMα1 and PSMα4 have previously been described to form typical fibrillar amyloid structures when aggregated on their own, as opposed to PSMα2 and PSMα3, which form α-helix-rich fibrillar aggregates [35,36]. Indeed, in our hands, PSMα1 and PSMα4 regularly showed increases in ThT fluorescence, whereas PSMα2 and PSMα3 did not (Figure 1 and Appendix A). As such, at higher peptide concentrations, the aggregation of PSMα1 and PSMα4 could potentially occur in parallel with α-syn aggregation. A retardation of α-syn aggregation can also be seen upon the addition of very high concentrations of lipid vesicles [37]. This is thought to be due to the adsorption of α-syn monomer to the membrane, effectively lowering the free monomeric α-syn concentration. Thus, the reduced catalytic effect of α-syn aggregation at high concentration of PSMα1 or PSMα4, but not PSMα2 or PSMα3, could indicate different affinities for α-syn to each of the PSMα peptides.

### 2.4. Composition of Aggregated Species

When the interactions between α-syn and the different PSMα peptides were probed by peptide arrays, α-syn did not show any signal above cutoff for the arrays with PSMα1 or PSMα4 peptides. However, α-syn exhibited a strong signal, indicative of high affinity binding to the PSMα2 sequences IIAGIIKFIK, GIIKFIKGLI, and the PSMα3 sequence FVAKLFKFFK (Appendix A).

Mass spectrometry was used to gain insight into whether the PSMα peptides formed low molecular weight heteromolecular aggregates with α-syn. Different PSMα-α-syn samples after reaching the plateau phase in the aggregation process were separated by SDS PAGE and the <10, 37, and 120 kDa bands were excised, subjected to in-gel digestion and analyzed with mass spectrometry. PSMα sequences were detected only in the lowest Mw bands (<10 kDa). All other bands showed the presence of α-syn but not PSMα (Appendix A). We thus hypothesize that the PSMαs induce α-syn aggregation by transient interactions but do not become incorporated in co-aggregates.

### 2.5. PSMα Peptides Induce α-Syn Aggregation by Heterogeneous Primary Nucleation

Although the data for PSMα-induced α-syn aggregation displaying significant deviation between repeats, models of increasing complexity were used to find out how many microscopic steps are needed to capture the observed behavior. For simplicity, these analyses ignored the biphasic curves observed at high concentration of some of the PSMαs. Clearly, the simplest model of primary nucleation and elongation predicts more shallow curves than those observed (Figure 2 and Appendix A). The very steep transition following a relatively flat lag phase requires a model that incorporates a secondary step, i.e., a step involving α-syn fibrils. A model of primary nucleation, elongation and secondary nucleation can reproduce most of the data in terms of curve shape and concentration dependence using a globally fitted value of the combined rate constant for secondary pathway (k_+_k_2_), while the combined rate constant for primary pathway (k_+_k_n_) was allowed to take different values at the different PSMα concentrations (Appendix A). The catalysis observed in the presence of PSMα peptides can thus be explained by the peptides inducing α-syn aggregation through heterogeneous primary nucleation, whereas the secondary nucleation step is intrinsic to α-syn.

### 2.6. α-Syn Concentration Dependence Varies for Different PSMα Peptides

To examine the α-syn-monomer-concentration-dependence of the observed catalysis, the PSMα peptide concentration was kept constant and α-syn monomer concentration varied. For all peptides examined, a significant acceleration of aggregation of α-syn can again be observed (Figure 2 and Appendix A) compared to α-syn without the peptide, which did not show an increase of ThT signal. For the pure peptides at 9 μM total concentration, ThT is only seen to emit a signal in the presence of PSMα1. Interestingly, PSMα1 on its own, reaches the plateau faster than any of the PSMα1- α-syn co-incubated samples. (Appendix A). The aggregation curve of PSMα1 at 9 μM follows a typical sigmoidal curve, whereas at 78 μM a biphasic aggregation curve is observed (Appendix A and Figure 1C).

For α-syn aggregation induced by PSMα1, a weak monomer-concentration dependency was observed. This may imply that the number of catalytic sites for heterogenic primary nucleation are limiting, or that product release is the rate-limiting step [38].

For PSMα2, a biphasic aggregation curve was observed for all concentrations of α-syn, especially obvious at 9 and 26 μM. A stepwise macroscopic curve may result if two different species form separate aggregates [39], if the process becomes physically restricted, e.g., due to restricted diffusion from gel formation or accumulation of intermediates at interfaces (Appendix A) [40]. At high α-syn monomer concentrations, the aggregation is less catalyzed than at lower α-syn monomer concentrations. Indeed, from the previous experiment, it can also be observed that PSMα2-induced catalysis of α-syn aggregation is less effective when the monomer to peptide concentration exceeds a ratio of 1:1.

Even though PSMα3 did not induce aggregation of α-syn at monomer concentrations below 26 µM (Appendix A), the aggregation shows a tendency to slower aggregation at higher monomer concentration. α-syn aggregated in the presence of PSMα4 (Appendix A), on the other hand, exhibited the fastest aggregation at the highest and lowest monomer concentration, compared to the mid-range concentrations.

The peptides are all potent inducers of α-syn aggregation, likely by heterogenous primary nucleation.

### 2.7. PSMα Peptides Induce α-Syn Fibril Formation

To examine the composition of the formed aggregates, the PSMα peptides and α-syn incubated together or separately were immunogold-labelled using an anti-α-syn antibody and imaged by TEM. PSMα-induced α-syn aggregates were found to be fibrillar and of approximately 10–20 nm in diameter (Figure 3 and Appendix A). α-syn incubated on its own occasionally showed individual fibrils despite the lack of increase in ThT signal in the kinetics experiments, in agreement with the formation of significant amounts of fibrils during the lag phase, although their concentrations need to reach about 1% of the final value to be detected by ThT bulk assay [41,42]. α-syn incubated on its own did, however, show higher background immunogold labeling, indicating high monomeric α-syn content. All the PSMα peptides appeared as amorphous aggregates.

### 2.8. PSMα-Peptide-Induced-α-Syn Aggregates Seed α-Syn in Cells

To investigate whether the formed α-syn aggregates could act as seeds in cells, the aggregates formed in α-syn-PSMα mixtures were added to HEK 293T cells expressing the A53T mutant form of α-syn fused to GFP. The A53T mutant form was used as it results in a robust aggregation model [43]. After 48 h, the cells were fixed and stained for phosphorylated α-syn. All imaged wells showed some signal indicating aggregate formation in the cells. The quantification of the GFP aggregates is less accurate than the quantification of the phosphorylated aggregates owing to the lower signal-to-noise ratio. The aggregates formed in different PSMα-α-syn mixtures showed differences in their ability to induce phosphorylation of aggregates in cells (Figure 3 and Appendix A). α-syn aggregated in the presence of PSMα1 and PSMα4 showed the highest phosphorylation induction at a 3:1 protein to peptide concentration ratio, whereas for PSMα2 and PSMα3, the highest phosphorylation induction was observed for protein aggregated at a 1:1 peptide concentration ratio. This is in agreement with the kinetic data, where PSMα1 and PSMα4 induce rapid α-syn aggregation at lower concentrations than PSMα2 and PSMα3. This indicates that the concentration of PSMα affects the seeding potential of the formed aggregates. When cells were treated with α-syn incubated in the absence of PSMα, no phosphorylated aggregates could be seen.

## 3. Discussion

α-syn exists as an intrinsically disordered monomer in vitro, and forms aggregates in vivo under disease conditions as well as in buffer systems in vitro. After the formation of the initial aggregates, rapid amplification of the aggregate mass can occur by autocatalytic secondary nucleation or by monomer addition to existing aggregates in an elongation process [44]. However, formation of the initial seeds (de novo seed formation) in bulk solution is slow and surface catalysis seems critical. The energy barrier for homogenous primary nucleation, i.e., from monomer species in solution, is high and primary nucleation is undetectable for α-syn at neutral pH, at least over a typical experimental time-frame of a few weeks [45]. Heterogenous primary nucleation, on the other hand, occurs on the surface of other substances, such as polystyrene plates or nanoparticles [46] or lipid vesicles [37], or at the air-liquid interface [47]. In vivo, primary nucleation is thought to be mainly heterogenous [44]. Extrinsic inducers of aggregation would explain the role of the environment in idiopathic PD. In our current work, we identified the bacterial amyloids, formylated PSMα peptides produced by *S. aureus,* as potent catalyzers of α-syn aggregation.

The α-syn aggregation mechanism is highly dependent on solution conditions such as pH and ionic strength, owing to the heterogenous charge distribution of the protein [45]. At a mildly acidic pH, the overall net negative charge of α-syn is reduced in contrast to a neutral pH, leading to a decrease in electrostatic repulsion between α-syn monomers, and between monomers and fibrils, and an increase in aggregation propensity. High salt concentrations, on the other hand, have been shown to screen electrostatic attractions between the oppositely charged termini within or between monomers in solution and on surfaces, resulting in a retardation of aggregation [48]. In this study we show that the PSMα peptides rapidly induce α-syn aggregation even at neutral pH and moderate ionic strength. All the examined PSMα peptides accelerate α-syn aggregation, although the concentration-dependence of these effects seems to differ. It is possible to explain most of the observed data, ignoring the biphasic curves observed in some concentration regimes, through the catalysis of α-syn aggregation by primary heterogeneous nucleation with the rate of nucleation being dependent on the ratio of PSMα peptide to α-syn. As illustrated in Figure 4, cross-catalysis and heterogeneous primary nucleation could, for example, occur through the formation of joint oligomers or on the surface of fibrillar forms of the peptides. The PSMα peptides themselves have been shown to have different pH and concentration dependent aggregation kinetics [49]; their different effects on α-syn have been shown in this study. Further work may elucidate the molecular origin of the cross-catalytic effects and how it varies among the PSMα peptides.

Peripheral α-syn expression is well documented and several studies have shown α-syn propagation and seeding from the periphery to the CNS [3,4,5,6]. Thus, we studied whether the PSMα-induced α-syn aggregates could induce seeding in cells from a cell-line. Indeed, upon direct addition to cells, the aggregates formed in α-syn-PSMα mixtures catalyzed α-syn aggregation in the cells and increased phosphorylation of the aggregates. Again, we could see differences in this catalytic effect, depending on which PSMα was present and its concentration. Likewise, in the aggregation kinetics, we have seen a clear dependence of the aggregation rate on the ratio of peptide to α-syn. This implies that the aggregates formed in α-syn-PSMα mixtures are potent inducers of α-syn aggregation and phosphorylation in cells.

Our current work identifies PSMα peptides as potent extrinsic inducers of α-syn aggregation and potential triggers of α-syn pathology. Our current work also highlights the concentration-dependent interactions of the PSMα peptides and α-syn monomer on the aggregation kinetics and the triggering of aggregation in cells. This supports that underly genetic predispositions and bacterial load could affect the pathological outcome. Previous studies have shown propagation of aggregated α-syn from peripheral tissues to the CNS. Our work identifies a mechanism by which the original seeds of α-syn could arise in the periphery. Figure 4C illustrates a potential mechanism whereby an *S. aureus* infection could trigger a cascade of NETosis and increased PSMα production catalyzing α-syn aggregation in peripheral tissues. *S. aureus* is also a commensal bacterium, colonizing an estimated 30% of the population. Although the PSMα peptides contribute to biofilm formation, it is still debated whether *S. aureus* biofilm formation is present in non-infection scenarios [50]; to our knowledge, no study has yet examined the expression of these peptides in commensal situations. With S. *aureus* infections continuing to rise globally, with the elderly population especially affected, host–pathogen interactions will gain further interest from the research and medical fields. The current results have significant implications for the understanding of *S. aureus* in initiation of α-syn pathology in different synucleinopathies, such as PD.

## 4. Materials and Methods

### 4.1. α-Syn Production and Purification

Human wildtype α-syn was expressed and purified using heat treatment, ion exchange and gel filtration chromatography, as previously described [51]. α-Syn samples were aliquoted and stored at −20 °C.

### 4.2. PSMα Production

The kinetics experiments were performed by two different labs involved in this manuscript.

The four different PSMα peptides were produced by chemical synthesis and isolated using reversed-phase HPLC to 95% purity (synthesis and purification by the manufacturer) with the following sequences:

PSMα1: Formyl-MGIIAGIIKVIKSLIEQFTGK

PSMα2: Formyl-MGIIAGIIKFIKGLIEKFTGK

PSMα3: Formyl-MEFVAKLFKFFKDLLGKFLGNN

PSMα4: Formyl-MAIVGTIIKIIKAIIDIFAK

For the experiments described in Appendix A the peptides were purchased from GenScript Biotech, Netherlands, and for all other experiments, they were purchased from EMC microcollections GmbH as lyophilized peptides.

The lyophilized peptides were stored at −80 °C and used without further purification in all experiments except one. For one experiment, PSMα2 was further purified using size exclusion chromatography. The peptide was dissolved at 1.25 mg/mL in 6 M GuHCl, 10 mM MES and purified using a Superdex Peptide 10/300 GL column in 10 mM Tris, 50 mM NaCl, pH 7.6, with absorbance measurements at 214 nm and 256 nm.

### 4.3. Aggregation Kinetics Experiments

In order to obtain reproducible data, monomeric α-syn was isolated immediately prior to each experiment. Purified α-syn was lyophilized, dissolved in 6 M GuHCl,10 mM MES, pH 5.5 and purified using size exclusion chromatography into 10 mM MES, pH 5.5 or 10 mM Tris, 50 mM NaCl, pH 7.6 at the start of every experiment. Buffer solutions were filtered and degassed prior to each run. The purification was monitored by UV absorbance at 280 nm and only the central monomer peak collected. The absorbance of the collected α-syn fraction was measured at 280 nm and an extinction coefficient of 5960 M^−1^ cm^−1^ was used to calculate the concentration. Purified α-syn monomer was kept on ice to prevent aggregation.

The PSMα peptides were dissolved in dimethyl sulfoxide (DMSO) to a concentration of 2 mg/mL and then diluted in buffer. The final DMSO concentration ranged from 0.1 to 8.9% *v*/*v*. Complementary α-syn aggregation experiments without PSMα peptides were performed at the same DMSO concentrations.

To follow the aggregation process, samples were aliquoted into 96-well black Corning polystyrene half-area microtiter plates with PEGylated surface (Corning 3881) in the presence of 20 μM ThT and 0.01% NaN_3_. Replicates without ThT were also prepared for additions to cell cultures. For the first round of experiments, the α-syn concentration was kept constant at 25 μM and PSMαs added at 5 different final concentrations: 1, 3, 9, 26 and 78 μM. Plates were incubated under quiescent conditions at 37 °C and the fluorescence was measured for up to 400 h (excitation filter 440 nm and emission filter 480 nm). Each experiment was repeated 1–3 times. These experiments were performed in 10 mM Tris, 50 mM NaCl, 20 μM ThT, 0.01% NaN_3_, pH 7.6 or in 10 mM MES, 20 μM ThT, 0.01% NaN_3_, pH 5.5. Data from one experiment, representative of all, are shown in the results section and Appendix A.

For the second round of experiments, the PSMα concentrations were kept constant at 8.6 μM and α-syn added at concentrations of 1, 3, 9, 26, 78 and 116 μM. These experiments were performed in 10 mM Tris, 50 mM NaCl, pH 7.6 in the presence of 20 μM ThT and 0.01% NaN_3_.

To verify the findings, kinetic experiments were also performed at different solution and surface conditions. For these experiments, described in Appendix A, the PSMα peptides were dissolved in DMSO to a final concentration of 10 mg/mL and then diluted with milliQ water to a concentration of 2 mg/mL. Then, they were further diluted into PBS containing 40 µM ThT and 50 µM α-syn at a final volume of 150 µL in a 96 well black polystyrene microtiter plates. The dilutions were supplemented with DMSO to a final concentration of 5 % (*v*/*v*). In these plates, the aggregation of α-syn was triggered by heterogeneous primary nucleation at the polystyrene surface [46]. The ThT fluorescence was monitored under quiescent conditions at 37 °C and averages of 2 replicates were plotted.

### 4.4. Kinetics Analysis

The data sets from each experiment were analyzed using Amylofit [52]. The ThT curves were normalized and the half time, defined as the point of time at which the ThT fluorescence has reached half-way in between the starting baseline and ending plateau, was extracted. For α-syn incubated on its own or in the presence of DMSO, the ThT curves were normalized to the values of α-syn aggregated in the presence of the highest concentrations of PSMα from the same experiment. For PSMα2 and PSMα3 peptides incubated on their own, as the values did not increase, the ThT curves were normalized relative to the values of α-syn aggregated in the presence of the highest concentrations of the respective PSMα from the same plate. For the experiment shown in Appendix A, the ThT curves for fraction 1 and fraction 7 as well as α-syn only were normalized relative to the values from fraction 4. The averages and standard deviations of the half-times of 3 or 4 replicates for each condition were plotted against the concentrations of PSMα peptides or α-syn. Models of primary nucleation and elongation only or models also accounting for secondary nucleation were fitted to the kinetics data from the aggregation of α-syn in the presence of varying concentrations of the different PSMα peptides.

### 4.5. Electrophoresis and Mass Spectrometry

The samples were analyzed by SDS-PAGE using Novex 10–20% Tricine precast gels and Tris/Tricine SDS Running buffer (100 mM Tris base; 100 mM Tricine; 0.1% SDS; pH = 8.3). 20 µL of each sample were mixed with 5 µL of gel loading buffer (1.66 M Tris; 44.4% glycerol; 9.3% SDS; 9.3% ß-mercaptoethanol; 0.093% Coomassie Blue). 10 µL of each mixture were loaded, and the gel was run at 70 V for 15 min, followed by 120 V for 1 h. The gel was stained with InstantBlue^TM^ overnight for optimal band detection. Gel bands of interest were excised and subjected to in-gel digestion. Gel pieces (1 × 1 mm^2^) were washed twice in 50 mM ammonium bicarbonate (NH_4_HCO_3_)/50% acetonitrile (ACN), followed by dehydrated in 100% ACN. Digestion was performed by adding 50 mM NH_4_HCO_3_ with 12 ng/µL sequencing-grade modified trypsin (Promega, Madison, WI, USA), incubation on ice for 4 h before overnight incubation at 37 °C. The next day trifluoroacetic acid (TFA) was added to a final concentration of 0.5% and the peptide containing solution above the gel pieces was withdrawn and used for further analysis. Mass spectrometry analysis was performed in reflector positive mode on an Autoflex Speed MALDI TOF/TOF mass spectrometer (Bruker Daltonics, Bremen, Germany). All peptide mass spectra were externally calibrated using Bruker Peptide Calibration Standard II.

### 4.6. Peptide Array

To probe interactions between α-syn and the PSMαs, 48 different 10-residue peptides corresponding to different parts of the PSM sequences were immobilized on a microarray chip and incubated with Alexa-fluor488-labelled α-syn. In this array, each new peptide constituted a 10-residue window of a given PSM sequence shifted forward by 1 residue compared to the preceding peptide, giving a 9-residue overlap. 

As part of this procedure, the microarray was first blocked in a solution containing 3% (*w*/*v*) skim milk in PBS with 0.1 % Tween-20 incubated overnight at 4 °C and washed three times with 25mM TBS with 0.1% Tween-20 (TSB-T). Subsequently it was incubated with Alexa-fluor488-labelled α-syn (diluted to 0.5 mg/mL in PBS) for 4 h at room temperature. The microarray was washed 3 times with TSB-T, air-dried in the dark and scanned using a Typhoon Trio scanner (GE Life Sciences, Pittsburgh, PA, USA). Dot intensities in the scanned image were quantified using ImageJ. 

### 4.7. Electron Microscopy

Samples were diluted 1:500 in sterile phosphate buffered saline (HyClone, 10126473, Utah, UT, USA) and 5 µL diluted sample added onto carbon-coated grids (made in-house). After washing in distilled H_2_O, the surface was blocked with bovine serum albumin (Sigma, A2153, Missouri, MO, USA) to prevent nonspecific binding and the specimens then incubated with 1:200 syn211 (sc-12767) for 1 h. Samples were again washed in distilled H_2_O, incubated with 1:20 10 nm gold-anti-mouse antibody (Ted Pella, 15751, California, CA, USA), washed, fixed first with glutaraldehyde (Ted Pella 18427) and then uranyl acetate (Agar Scientific R1260A, Essex, UK). The samples then were examined under a transmission electron microscope (TEM) (FEI Tecnai Biotwin 120 kV, Minnesota, MN, USA).

### 4.8. HEK 293T Culture

HEK 293T cells were stably transduced with the A53T mutant of α-syn fused to GFP and were maintained in DMEM GlutaMAX (Gibco 10569010, Texas, TX, USA) supplemented with 10% fetal bovine serum (Gibco 10270-106) and 1% penicillin-streptomycin (Gibco 15140-122) and split every three days.

For live-imaging experiments, cells were split 1:20,000 into collagen G-coated (Biochrom GmbH L7213, Berlin, Germany) black, clear bottom 96-well plates. Samples of α-syn and PSMα incubated separately or together were sonicated at 100% amplitude for 3 min at 50% duty cycle (1 s on, 1 s off). 10 µL of each sample was added in 4 or 3 replicates into 100 µL medium. The plates were imaged for 48h with 1h intervals using a Nikon Ti-E microscope with 95% humidity 5% CO_2_.

### 4.9. Phospho-α-Syn Staining of HEK 293T Cells

After 48 h, the cells were fixed with phosphate-buffered paraformaldehyde pH 7.4 by slow addition into the cell medium to a final concentration of 2% for 20 min. The cells were then washed and kept at 4 °C. Cells were then permeabilized and blocked with 5% normal donkey serum (NDS) 0.25% Tween 20 in PBS. Primary antibody pS129 α-syn (Abcam 51253, Cambridge, UK) was added at 1:1000 overnight and the plates kept at 4 °C. The cells were washed and incubated with 1:600 Cy3 conjugated anti-rabbit antibody (Jackson Immunolabs 711-165-152), washed and then imaged again. For each well, 25 (5 × 5 square) images were acquired with a 20× objective.

### 4.10. Quantification of Aggregates

Quantification of the number of GFP-aggregates and pS129 α-syn inclusions was performed using Cell Profiler. In brief, cell nuclei were identified as primary objects by segmentation by diameter sizes and counted in the GFP channel with a global thresholding strategy. In the GFP channel and the Cy-3 channel, aggregates and phosphorylated aggregates, respectively, were identified as secondary objects by segmentation by diameter size and counted using an Otsu adaptive thresholding strategy, which separates foreground from background, optimally used when percentage of the image covered by foreground varies substantially from image to image. The number of aggregates and cells were quantified per well. The number of aggregates and phosphorylated aggregates per 100 cells were determined per well and averages from 4 replicates were plotted for each condition.

## Figures and Tables

**Figure 1 ijms-22-11594-f001:**
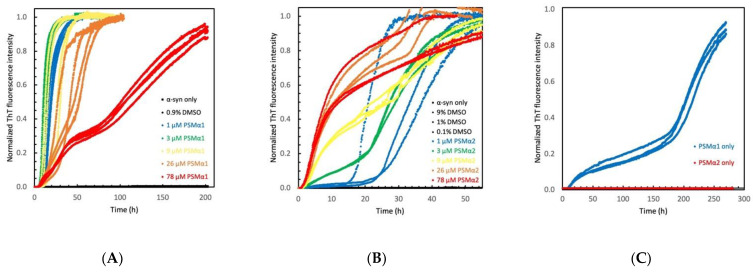
Aggregation of α-syn in the presence of PSMα peptides. (**A**,**B**) Normalized ThT fluorescence intensity as a function of time for 25 µM α-syn in the presence of varying PSMα1 (**A**) or PSMα2 (**B**) concentrations as provided in the respective panel. (**C**) Control experiments with only 78 µM PSMα1 (blue) or 78 µM PSMα2(red). All experiments started from monomers and were performed in 10 mM Tris, 50 mM NaCl, pH 7.6 at 37 °C under quiescent conditions.

**Figure 2 ijms-22-11594-f002:**
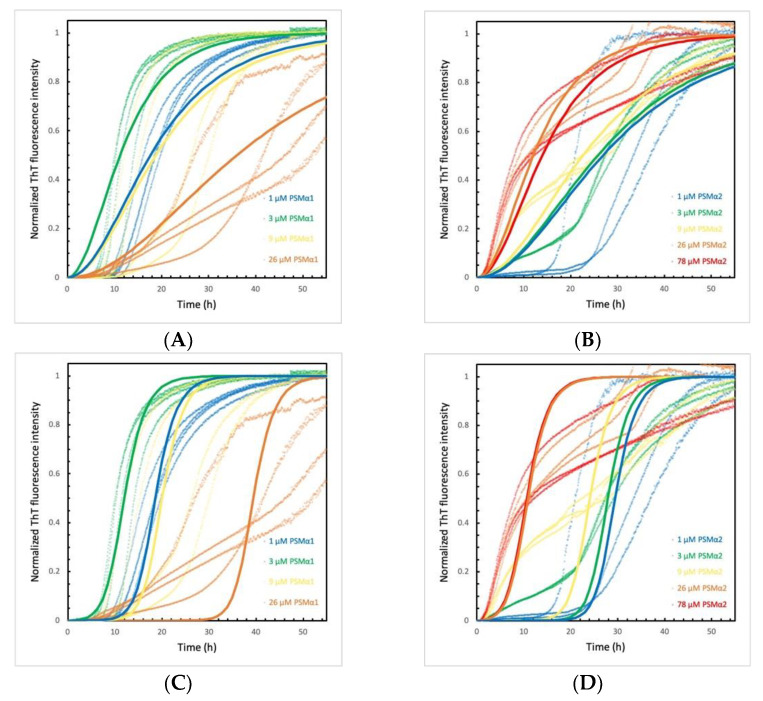
Fitting of models of nucleation and elongation to aggregation kinetics of α-syn in the presence of PSMα peptides. The ThT fluorescence intensities as a function of time of 25 µM α-syn aggregation induced by varying PSMα1 (**A**,**C**) and PSMα2 (**B**,**D**) concentrations (µM) were fitted to models of primary nucleation and elongation (**A**,**B**) or to models of primary nucleation, elongation and secondary nucleation (**C**,**D**). The dotted lines are the normalized ThT fluorescence intensities and the solid lines are the fits as calculated by Amylofit. All experiments started from monomers and were performed in 10 mM Tris, 50 mM NaCl, pH 7.6 at 37 °C under quiescent conditions.

**Figure 3 ijms-22-11594-f003:**
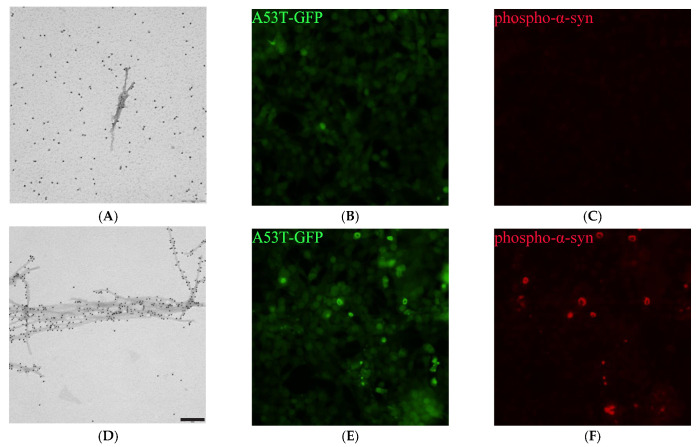
Microscopic images. (**A**,**B**) TEM micrographs at 60,000 times magnification of samples taken at the plateau of aggregation reactions with 25 µM α-syn (**A**) in the absence or (**B**) presence of 9 µM PSMα1 peptide. All experiments started from monomers and were performed in 10 mM Tris, 50 mM NaCl, 1.1% DMSO, pH 7.6 at 37 °C under quiescent conditions. After deposition on the grids, the samples were labelled by an anti- α-syn primary antibody (syn211) and a secondary antibody linked to 10 nm gold nano-particles. Scale bar = 200 nm. (**C**–**F**). GFP and Cy-3 fluorescence microscopic images at 20 times magnification of HEK cells expressing A53T α-syn fused to GFP. The same samples as imaged by TEM were sonicated and added at a 1:10 ratio to the cells for 48 h before the cells were fixed and incubated with an antibody against phosphorylated α-syn followed by a Cy3-conjugated secondary antibody.

**Figure 4 ijms-22-11594-f004:**
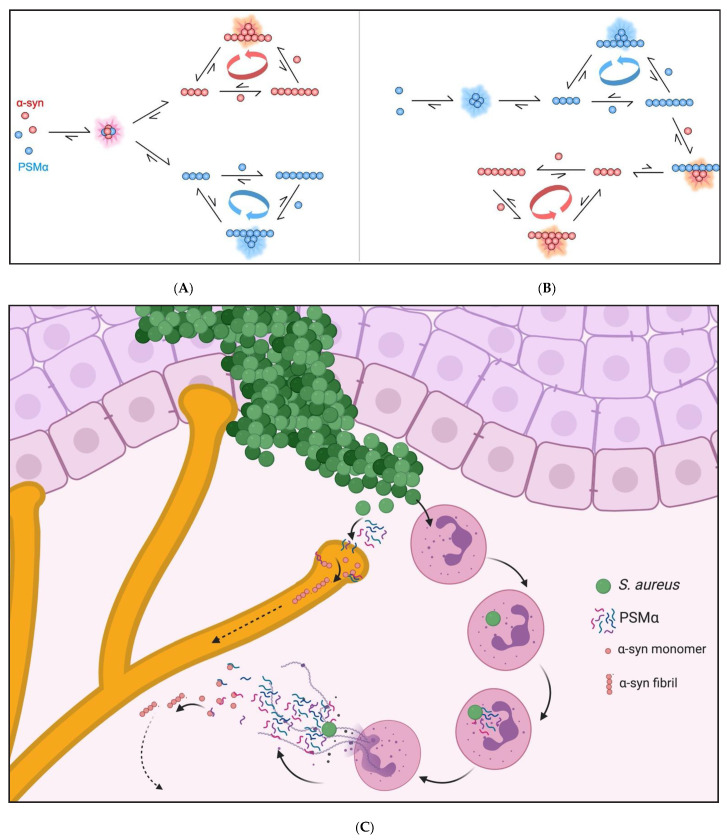
Schematic illustration of outlining possible modes of PSMα-induced α-syn aggregation. Cartoon illustrating two of many possible modes of cross-catalysis between PSMα (blue) and α-syn (red) involving heterogeneous primary nucleation. The curved arrows indicate the autocatalytic cycles that may result from homogenous secondary nucleation of each peptide on the fibrils of the same peptide. (**A**) Cross catalysis occurs via heterogeneous primary nucleation in joint oligomers, with no cross-catalysis at fibril level. (**B**) Separate primary nucleation, significant only for PSMα, and heterogeneous primary nucleation of α-syn on PSMα fibrils. (**C**) Cartoon illustrating possible routes by which *S. aureus* and PSMαs could lead to α-syn pathology. Invading *S. aureus* produces PSMα peptides that are chemoattracting neutrophils. Neutrophils phagocytose *S. aureus* bacteria, whereby PSMα expression is upregulated leading to phagosome escape and NETosis (24, 25, 27). At high concentrations, the PSMα peptides may interact with extracellular α-syn causing aggregation by enhanced primary nucleation, leading to the formation of fibrils, which can propagate and seed α-syn in other cells (this study). PSMα peptides also form pores in sensory neurons (28, 29). The PSMα peptides may then enter cells and interact with intracellular α-syn.

## Data Availability

All raw data is available upon reasonable request.

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
