# Peer review of "The Bacterial Amyloids Phenol Soluble Modulins from Staphylococcus aureus Catalyze Alpha-Synuclein Aggregation"

_ijms, 2021, doi:10.3390/ijms222111594_

Round 1

Reviewer 1 Report

The manuscript interestingly describes the effect of PMSalpha peptides on alpha-synuclein aggregation. Findings are of great interest considering the suggested role of microbiome in PD.

However, findings in this report are very very confusely described in many points and the text is really difficult to follow.

Page 3 line 114: the authors stated that 1-100 uM range of PSMalpha-s are physiologically relevant. Why? cite references about this point and explain the choice of those doses

Page 3 line 124: why 25 uM alpha-synuclein concentration was used for experiments? please explain the choice

Page 4 line 151: why PSMalpha4 was not tested in suppl fig 3?

Page 4 line 160: I do not understand why dmso concentration increased with peptide concentration (in these type of experiments the vehicle concentration is usually maintained the same), please explain

Page 5 lines180-183, please rephrase, the text is unclear

Page 5 suppl fig 6: the legend of this figure is very confusing, please explain better in the legend and in the text the results of the experiments

Page 5, line 185 and Figure 3: in these experiments A53T alpha-synuclein is used, why not wild type protein as in prior and further experiments?? I really do not understand

Minor points:

page 2 line 76: reference 41 is cited after ref 21 and before ref 22

Author Response

Reviewer #1:

The manuscript interestingly describes the effect of PMSalpha peptides on alpha-synuclein aggregation. Findings are of great interest considering the suggested role of microbiome in PD.

However, findings in this report are very very confusely described in many points and the text is really difficult to follow.

  1. Page 3 line 114: the authors stated that 1-100 uM range of PSMalpha-s are physiologically relevant. Why? cite references about this point and explain the choice of those doses
    1. Thank you for comments. Based on references cited in the introduction (23 and 24) high nanomolar and micromolar concentrations of these peptides can attract neutrophils, penetrate the plasma membrane and cause cytolysis. According to reference 29, 10 µM can induce firing of cultured DRG neurons and 5-10 nmol could induce spontaneous pain. As such, we chose to study this range of concentrations.
  2. Page 3 line 124: why 25 uM alpha-synuclein concentration was used for experiments? please explain the choice
    1. As can be seen in Supplementary Figure 10, we have tested a range of different a-syn concentrations. At very low concentrations, the signal to noise ratio is quite low. 26 µM produced robust kinetics within the timeframe of our experiments.
  3. Page 4 line 151: why PSMalpha4 was not tested in suppl fig 3?
    1. We have now added texts to explain that these experiments were carried out by two different labs involved in this manuscript (initially measured by the Li lab in collaboration with the Linse lab and subsequently by the Otzen lab). Due to constraints associated with peptide synthesis, the second lab had access to only 3 of the peptides. However, we find that to validate our finding that the peptides catalyze a-syn aggregation, the presented data is sufficient.
  4. Page 4 line 160: I do not understand why dmso concentration increased with peptide concentration (in these type of experiments the vehicle concentration is usually maintained the same), please explain
    1. We thank the reviewer for this comment and have now improved our explanation. On lines 172-174 we now explain as such: “As the PSMα-α-syn aggregated samples would later be added to cells, the lower concentrations of the peptides were not supplemented with additional DMSO.” We have also included a control experiments where the PSMa2 concentration varied, but DMSO did not, as shown in Supplementary Figure 4.
  5. Page 5 lines180-183, please rephrase, the text is unclear
    1. We have amended this. In lines 393-395 we state: “To examine the composition of the formed aggregates, the PSMα peptides and α-syn incubated together or separately were immunogold-labelled using an anti-α-syn antibody and imaged by TEM.”
  6. Page 5 suppl fig 6: the legend of this figure is very confusing, please explain better in the legend and in the text the results of the experiments
    1. Thank you for drawing our attention to this. We have re-organized the data to better explain our results and rephrased our findings (see lines 392-402): “To examine the composition of the formed aggregates, the PSMα peptides and α-syn incubated together or separately were immunogold-labelled using an anti-α-syn antibody and imaged by TEM. ”
  7. Page 5, line 185 and Figure 3: in these experiments A53T alpha-synuclein is used, why not wild type protein as in prior and further experiments?? I really do not understand
    1. Thank you for your question on this issue. Previous studies have shown that although both wildtype and mutant (A53T) a-synuclein can get aggregated in vitro and in vivo models, the A53T a-syn possess higher aggregation capacity. We therefore have chosen to use the A53T mutant in our experiments, since they are performed during a relatively short timespan. We have now included this information in the texts (line 408-409) and added a reference: 43.
  8. Minor points:
  9. page 2 line 76: reference 41 is cited after ref 21 and before ref 22

Reviewer 2 Report

This study and findings represent an interesting area of relevance to linking gut microbiota with synuclein pathology. However the manuscript in its current format is extremely difficult to read and follow which would need to be addressed before it could be considered for publication. More details on the current issues are provided in the attached document.

Author Response

Reviewer #2:

  1. Data for peptides alone at pH5.5 is not included (relevant to Supplementary Figure 2).
    1. Sorry for having missed this. We have now included these data figures
  2. It is hard to compare normalised asyn+peptide data with non-normalised peptide alone data e.g. Figure 1 and Supp Figure 1.
    1. We have solved this problem by normalizing all data
  3. When discussing monotonic and biphasic action of peptides only 1 and 2 are mentioned, what about 3 and 4?
    1. Thank you for commenting on this. We have included additional information to the texts regarding peptides 3 and 4 now.
  4. The authors exclude molecular crowding based on the data presented but I do not feel that the experiments showed in the manuscript can exclude molecular crowding as they haven’t included a control of another peptide of similar size but without catalysing activity?
    1. We appreciate your comment and agree with you on this important issue. We have explained this in the text now (lines 145-147). Had this been an effect of molecular crowding we would have expected identical kinetics for all the peptides as they are similar in size and charge.
  5. The authors state that peptide 3 only catalysed asyn aggregation at low concentrations at pH5.5 but Supp Figure 1 also shows asyn aggregation at pH7.6 in the presence of peptide 3?
    1. We thank the reviewer for pointing us to this error, we have amended it now.
  6. Why was peptide 4 excluded from analysis shown in Supp Figure 3? Also this figure would benefit from DMSO alone control and peptide alone controls. Why were different peptides used for data in this figure compared to the rest of the manuscript?
    1. We have now added texts to explain that these experiments were carried out by two different labs involved in this manuscript (initially measured by the Li lab in collaboration with the Linse lab and subsequently by the Otzen lab). Due to constraints associated with peptide synthesis, the second lab had access to only 3 of the peptides. However, we find that to validate our finding that the peptides catalyze a-syn aggregation, the presented data is sufficient. PSM only controls are included in reference 22, an article in which the Otzen lab has studied the effect of heparin on PSM peptides.
  7. Supp Figure 4, x axis in part B is labelled incorrectly. The description of this experiment is very confusing currently so should be made clearer. A direct comparison of data with 9% DMSO as shown in Supp Figure 4B with that shown in Figure 1 might help too.
    1. We have amended the x-axis label in supp fig 4 now.
  8. Figure 2 is also highly confusing and not discussed well or in the correct place within the text to enable the reader to follow the figures and text in order. Currently Figure 2A is referred to briefly before Figure 3 is discussed. Peptide alone should be included in B).
    1. We thank the reviewer for the input. We have now reorganized the data and for figure 2 instead show the fits from the amylofit models. We have placed all of the a-syn monomer concentration data in Supplementary Figure 10 and hope this makes the results clearer.
  9. Supp Figure 6 – why is part H a different style scale bar? The images shown in part E appear to contain fibrillar like species whereas part G do not. This does not fit with the ThT data for these corresponding peptides, can the authors comment on these data?
    1. We thank the reviewer for their comment, indeed it is confusing. We have removed the different scale bar. We had added that in case the original scale bars were too small to be seen. If looked at closely, there is no actual fibril in any of the PSM peptides alone. All of the images show amorphous aggregates.
  10. line 209-211 the authors reference Supp Figure 1 for peptide alone ThT, this figure does not contain data for all peptides.
    1. We have fixed this now.
  11. Supp Figure 8 legend needs to state that solid lines are AmyloFit data
    1. We thank the reviewer for drawing our attention to this, we have now fixed it.
  12. Supp Figure 9 legend should state what triangles in part B represent. The y axis scale makes it difficult to relate t1/2 values shown in E and F with ThT curves above.
    1. We have changed this figure now.
  13. Gel has lanes labelled 1-14 but legend has them 1-15.
    1. We have amended this.
  14. Supp Figure 12 shows high variability between different experiments which is not discussed and should be as the conclusions are correct that this data shows promise I do not feel it is as strong based on these graphs as the manuscript claims e.g. peptide 1 elicits a greater production of phosphorylated asyn than in the presence of asyn in one of the experiments and most peptides only exhibit a positive result in 1 out of 2 experiments.
    1. We appreciate the reviewer’s comment. For the cell data, in all cases, the cells treated with only asyn show fewer phosphorylated aggregates than cells treated with asyn aggregated in the presence of PSM peptide. Indeed, there is variability in the absolute numbers of aggregates identified, owing to slightly different imaging settings and automated quantification. Yes, PSMa1 alone shows an increased number of pSyn aggregates. The amount of peptide added is quite high compared to the peptide concentration in the asyn+ peptide conditions. These cells were incubated with the materials for 48h, meaning that if there were to be any extracellular a-syn from the cells, or if PSM was internalized, this asyn could have been catalyzed to aggregate within that time-frame.

Additional minor errors:

Line 257 needs the word of inserting. Some figures have strange formatting of text in the legend. Line 223 and 361 have incorrect spacing. Supp Figure 10 and Figure have typing errors. 

There is no consistency between the use of capital letters for the word Figure/figure and in figures/legends (A or a) e.g. Figure 1 legend has A and B in brackets but not for C. Figure 2 legend does not refer to parts correctly in the legend. There are several instances of words being hyphenated unnecessarily e.g. concen-trations, neutro-phils.

  1. Sorry for the inconsistency appeared. We have checked the whole manuscript carefully, have corrected all noted errors.

Round 2

Reviewer 1 Report

Thank you for the comments, I feel the manuscript is now acceptable for publication also considering the correct requests of the second reviewer and the comments of the authors.

Reviewer 2 Report

The changes to the manuscript have made it easier to follow and added in the relevant missing control data. My only remaining concern is regarding the strange presentation of the PSMa4 only curve in Supplementary Figure 2C?
